# Voluntary Oral Ingestion of a Sedative Prior to Euthanasia with CO_2_: Behavioural Responses of Mice

**DOI:** 10.3390/ani11102879

**Published:** 2021-10-01

**Authors:** Raquel Rodriguez-Sanchez, Elyssa Barnaby, Lucia Améndola, Shen-Yan Hea, Bobby Smith, James Webster, Gosia Zobel

**Affiliations:** 1Animal Behaviour and Welfare Team, AgResearch Ltd., Ruakura Research Centre, 10 Bisley Road, Private Bag 3123, 3214 Hamilton, New Zealand; Raquel.Rodriguez@agresearch.co.nz; 2Animal Ethics Office, AgResearch Ltd., Ruakura Research Centre, 10 Bisley Road, Private Bag 3123, 3214 Hamilton, New Zealand; Elyssa.Barnaby@agresearch.co.nz (E.B.); jim.webster@agresearch.co.nz (J.W.); 3Atlantic Veterinary College, University of Prince Edward Island, 550 University Ave, Charlottetown, PE C1A 4P3, Canada; luciamendola@gmail.com; 4Bioinformatics and Statistics, AgResearch Ltd., Grasslands Research Centre, Tennent Drive, Fitzherbert, 4410 Palmerston North, New Zealand; Shen.Hea@agresearch.co.nz; 5Small Animal Colony, AgResearch Ltd., Ruakura Research Centre, 10 Bisley Road, Private Bag 3123, 3214 Hamilton, New Zealand; bobby.smith@agresearch.co.nz

**Keywords:** welfare, C57BL/6 mouse, tiletamine, zolazepam, sedation

## Abstract

**Simple Summary:**

Carbon dioxide (CO_2_) inhalation is the most common euthanasia method for laboratory mice despite causing distress, pain and suffering. Since many laboratories will continue to use CO_2_ until a suitable alternative is identified, there is merit in exploring options for reducing its aversiveness. We evaluated the potential of using a voluntarily ingested sedative (tiletamine-zolazepam; Zoletil^®^) prior to euthanasia. Male and female C57BL/6 mice were offered cream cheese mixed with Zoletil^®^ in one of the following doses: 0, 10, 20, 40, 80, or 100 mg/kg. Behaviour during the sedation and euthanasia periods was recorded. A dose of 20 mg/kg was found to achieve mild sedation and was likely to reduce the aversiveness of euthanasia with CO_2_. Higher doses also produced sedation, but these resulted in an incomplete intake of the dosed cream cheese in some mice. The voluntary ingestion of a sedative (20 mg/kg) prior to CO_2_ deployment could be a valid option for reducing the stress of this euthanasia method for both mice and staff. Additionally, we suggest that since mice readily ingested dosed cream cheese, this could also be an easy, effective, non-invasive, and low-cost means of reducing stress in other applications (e.g., repeated handling or sampling).

**Abstract:**

Laboratory mice are commonly euthanised with carbon dioxide (CO_2_); however, there is ample evidence that this gas is aversive. Previous work suggests that sedation achieved via injection with benzodiazepines prior to CO_2_ administration could reduce aversive behaviours during euthanasia. We explored the potential of using a voluntarily ingested sedative (tiletamine-zolazepam, Zoletil^®^) prior to euthanasia. Male and female C57BL/6 mice were allocated into one of the five experimental groups, which differed in the dose of Zoletil: 0, 10, 20, 40, 80 or 100 mg/kg. A dose of 20 mg/kg was found to achieve mild sedation prior to euthanasia; mice which received this dose numerically reared and walked on the cage lid less, and showed ataxia, immobility and recumbency for longer than mice that received a lower dose. During euthanasia, mice that received 20 mg/kg showed fewer aversive responses to CO_2_. Doses of 40 to 100 mg/kg were associated with signs of moderate to severe sedation, but resulted in an incomplete intake of the sedative, which made the interpretation of the aversiveness to CO_2_ difficult. Voluntary oral administration of a sedative is an effective, affordable, and easy way to minimize the stress of mice to euthanasia with CO_2_.

## 1. Introduction

Inhalation of carbon dioxide (CO_2_), delivered using a gradual fill protocol, is a common euthanasia method for laboratory mice [1,2,3]. CO_2_ is readily available, induces loss of consciousness in a rapid and reliable way with minimal personnel safety concerns and it has a relatively low cost [2,4,5]. However, CO_2_ is aversive for mice and likely causes considerable distress, pain, and suffering [2,6]; its use has been increasingly questioned [7,8,9]. In New Zealand, the National Animal Ethics Advisory Committee has taken the stance that CO_2_ euthanasia is outdated, and thus replacement euthanasia methodologies should be pursued [10]; indeed, the methodology has been questioned globally. Therefore, numerous alternative inhalants (e.g., argon, carbon monoxide, halothane, isoflurane, nitrogen) have been explored; however, these have been found to produce aversive or fear responses as well [6,11,12]. Since most laboratories will continue to use CO_2_ until a better alternative is identified, there is merit in exploring options for reducing its aversiveness.

Thomas et al. [13] suggested that combining CO_2_ with nitrous oxide gas (N_2_O) reduced the duration of distress during euthanasia. No jumping and a lower rearing frequency were observed once the gas flow started when CO_2_ was combined with N_2_O rather than when CO_2_ was used alone; however, the welfare impact of the lactic acidosis induced by N_2_O could not be measured. Spiacci et al. [14] explored the use of anxiolytics prior to euthanasia. The authors reported that intraperitoneal injection of diazepam did not reduce the number of escape attempts (vertical jumps indicative of aversion), while injection of alprazolam (0.1 mg/kg) did reduce the expression of this behaviour. Neither diazepam nor alprazolam affected locomotion before exposure to CO_2_, suggesting that these drugs did not have a sedative effect at the administered doses. Furthermore, in the case of alprazolam, the maximum dose administered reduced, but did not completely eliminate, the escape response. It is important to note that the handling required, and the injection itself, used in this method could be an aversive experience [15]. Therefore, we explored an alternative voluntary administration method that required minimal handling.

To our knowledge, voluntary ingestion of a sedative has not been considered as a precursor to CO_2_ euthanasia. Zoletil^®^ (Virbac, New Zealand), which is a combination of tiletamine (a dissociative anesthetic), and zolazepam (a benzodiazepine), can be used with other drugs (e.g., xylazine) as an anesthetic for mice [16,17,18]; typically, it is administered as an intramuscular, or intraperitoneal, injection. Methods of oral drug delivery have been described in mice to reduce stress, pain, and morbidity [19,20]. We explored the potential of using a voluntarily ingested sedative (tiletamine-zolazepam) prior to euthanasia. Mice habituate to, and readily consume, cream cheese; thus, we speculated that it could be a suitable medium in which to provide a sedative. Therefore, the objectives of this pilot study were to first determine the maximum dose of the sedative (tiletamine-zolazepam) that both male and female C57BL/6 mice could consistently consume, and then to assess the behavioural changes caused by the ingestion of this sedative prior to euthanasia with CO_2_. We hypothesised that prior sedation would reduce the expression of aversion-related behaviours known to be associated with the use of CO_2_ euthanasia (e.g., rearing, jumping), and decrease the time it would take for mice to become insensible (e.g., latency to recumbency).

## 2. Materials and Methods

### 2.1. Animals and Facilities

This study was undertaken at the Small Animal Colony at AgResearch’s Ruakura Research Centre (Hamilton, New Zealand) in October 2020. All procedures involving animals were approved by the Ruakura Animal Ethics Committee (AE14902) under the New Zealand Animal Welfare Act (1999). 

A total of 80 C57BL/6 mice (40 females, 40 males, aged 9 months, weighed 28.4 ± 4.1 g) were used in the study. Animals were housed in pairs of the same sex in polycarbonate cages (Techinplast 1144B; 331 mm L × 159 mm W × 132 mm H) under controlled temperature (22.4 ± 0.5 °C) and humidity (49.7 ± 7.4%) conditions, and under a 12 h light: 12 h dark cycle (lights on at 0700 h). Food (Rat and Mouse Cubes, Specialty Feeds, Australia) and tap water were provided *ad libitum*. Bedding material (Pura Paper, Able Scientific, Australia), nesting material (Pura Crinkle Paper, Able Scientific, Australia) and a nest box were provided in each cage. To differentiate between the two individuals in a cage, one mouse per cage was marked (ear punched). 

### 2.2. Experimental Design

An adaptive dose escalation design was used to investigate the sedative effects of increasing tiletamine-zolazepam (Zoletil^®^, Virbac, New Zealand) intake per os. This approach allowed the flexibility to adjust the next dose rate depending on the observations from the previous dose rate. 

A total of 5 dose escalation sessions were performed starting from an initial dose rate of 10 mg/kg. The dose rate for each subsequent session was doubled except for the final session when it became apparent that the increasing dose rates were affecting cream cheese intake (resulting in dose rates of 20 mg/kg, 40 mg/kg, 80 mg/kg, and 100 mg/kg). For each session, 8 mice from two male and two female cages were offered the dosed cream cheese (treatment cages). At each dosage session, each treatment cage was randomly paired with a control cage containing mice of the same sex, resulting in a total of 16 mice from four male cages and four female cages per session.

Mice were habituated to consume cream cheese (Meadow Fresh, New Zealand) for 5 consecutive days (day 1 to day 5) prior to euthanasia on day 6. Every day, each pair was temporarily split up in their home cage using a standard cage divider and each mouse was given 200 mg of cream cheese in a Petri dish. After 5 min, the cream cheese was removed and weighed to calculate a baseline quantity which would be used for dosing on day 6. The dividers were cleaned with soap and water and wiped with a dry paper tissue between each use and were autoclaved after each day. Mice were weighed on day 5 to allow for dose calculations.

On euthanasia day (day 6), the nesting material, food and water were removed from the cages for video recording purposes. Each pair of mice was temporarily split up in their home cage using the standard cage dividers used during habituation and provided with 150 mg of cream cheese either non-dosed (control groups) or containing a single dose (treatment groups) of sedative (tiletamine-zolazepam combination). The initial dose of sedative was determined as per Plumb’s Veterinary Drug Handbook (10 mg/kg IM was considered as a reference as there is no information available for PO doses in small rodents [21]); it was dissolved in water and the corresponding volume (depending on the dosing treatment) was mixed with the cream cheese. After 5 min, the cream cheese was removed and weighed. Mice were recorded for 20 min prior to the euthanasia (sedation period) to detect any signs of sedation. The cages were then moved one at a time from the housing room to a procedure room; mice were kept in their home cage, and the cage lid was removed and replaced with an acrylic plastic lid that had a CO_2_ inlet attached. The CO_2_ flow rate was 30–40% of the chamber volume per minute as recommended by the CCAC guidelines on euthanasia of animals used in science [22], and was turned off after 1.67 ± 0.31 min (mean ± SD) (euthanasia period). A veterinarian checked the animals to confirm death following euthanasia (cessation of heart rate and respiration, pupils fixed and dilated, lack of corneal reflex). The waiting time between the sedation period and the start of the euthanasia period ranged from 2 to 24 min (mean ± SD: 12.0 ± 5.5 min). 

### 2.3. Behavioural Observations

Mouse behaviour was monitored using GoPro cameras (GoPro Hero 7; GoPro Inc, San Mateo, CA, USA) directed at the short side of the cage for the sedation period and at the long side of the cage for the euthanasia period (behaviour was not recorded for the habituation period). Detailed behaviours were recorded for both sedation and euthanasia periods according to standardised ethogram (Table 1). During the sedation period, behaviours were originally recorded for 1200 s (20 min), but due to video loss for some mice, behavioural reporting was standardised to the first 800 s of this period. For the euthanasia period, behaviours were recorded until the mouse was deemed to have taken its last breath (ranging from 1.4 to 2.3 min following deployment of CO_2_; breathing slowed down to intermittent gasping and the recording continued for approximately 30 s after the last breath to confirm that it was).

A single-blinded observer watched all videos, with each mouse scored individually for all observations. After 5 observations, the observer randomly selected 1 of the videos to re-watch; this resulted in 20% of the observations being used to calculate intra-observer reliability (mean kappa: 0.98; range: 0.86–1.0).

### 2.4. Data Handling and Statistical Analysis 

Each mouse (within each cage) served as the experimental unit. For the sedation period, the duration(s) of ataxia, immobility, laboured breathing, paddling, recumbency, shaking and walking on lid were recorded. For the euthanasia period, latency to ataxia was recorded starting from the onset gas exposure (CO_2_); the remaining behaviours were then recorded as latency from the onset of the previous behaviour (e.g., latency to recumbency from the onset of ataxia, latency to laboured breathing from the onset of recumbency, time to last breath (death) from the onset of laboured breathing). Additionally, for both periods, frequency of rearing events and vertical jumps were recorded. One cage from the Control group and two cages from Z100 treatment could not be included in the behavioural analysis during the sedation period due to a camera recording issue. Before euthanasia, there were 4 mice in Z80 group and 2 mice in Z100 group that were already recumbent, and therefore they could not rear or jump. These animals could not be considered for the progression of events during the euthanasia because they were already recumbent when the euthanasia process started. Therefore, as the data were not missing at random, the results may be biased. Dose–response profiles were modelled using a 4-parameter log logistic model using the ‘drc’ package (analysis of dose–response curves) [26]. A linear model was used to describe the dose–response profile when the log logistic model parameters were unable to be estimated. For the analysis of rearing, the frequency of rearing was transformed into a binary variable (no rearing and rearing one or more times). A logistic regression model was used to assess the relationship between increasing dose rates and the probability of rearing. Results in tables are descriptive and are reported as least-squares means ± SD; minimum, maximum and quartile values are also included. Predicted plots include ±95% confidence interval. All statistical analysis and data manipulations were performed using the R software program [27].

## 3. Results

### 3.1. Habituation Period

Over 5 days, mice were habituated to consume cream cheese by providing it daily for 5 min. The latency to ingest the cream cheese was not recorded; however, by day 3, except for 2 females, all mice consumed more than 97% of the cream cheese offered. The two outlier females consumed approximately 50% on this day, and one (which was randomly allocated to Control group for the sedation period) never consumed more than 50% of the amount offered.

### 3.2. Sedation Period

We assessed the effects of tiletamine-zolazepam consumed voluntarily; therefore, the actual doses consumed by individual mice differed from the planned doses of tiletamine-zolazepam. Doses above 20 mg/kg (i.e., 40, 80 or 100 mg/kg) resulted in an incomplete intake of the cream cheese by 16 (10 male and 6 female) of the 24 mice. Males appeared to eat less than females especially in Z80 treatment, and therefore had lower doses of the sedative (Appendix A). The planned and actual doses of tiletamine-zolazepam for each group are presented in Table 2.

Behaviours observed during the sedation period are presented in Figure 1; these are reported according to the actual doses of tiletamine-zolazepam. Regardless of the treatment, vertical jumping, presence of laboured breathing, and paddling were infrequent (Figure 1a,g,h, respectively). Some vertical jumps were reported for Control, Z10 and Z20 treatments, and laboured breathing and paddling were detected in one mouse in Z80 treatment (Appendix A). 

A high rearing frequency was reported in both Control (112 ± 31.1 rears) and Z10 treatment (118 ± 28.4 rears) (Figure 1b); the rearing frequency declined as the tiletamine-zolazepam dose increased (Z20: 94 ± 42.9 rears, Z40: 74 ± 48.9 rears, Z80: 33 ± 47.9 rears, and Z100: 25 ± 18.6 rears; Appendix A). A similar pattern was observed for time spent walking on the lid (Figure 1f); while mice in Control, and Z10, Z20 and Z40 treatments walked on the lid, mice in Z80 and Z100 treatments spent minimal to no time preforming this behaviour (Control: 87 ± 63.8 s, Z10: 25 ± 17.8 s, Z20: 19 ± 29.5 s, Z40: 27 ± 46.5 s, Z80: 2 ± 4.4 s, Z100: 0 s; Appendix A). 

On the other hand, ataxia was never or rarely observed in Control and the Z10 treatment, respectively (Figure 1c), but it was reported in the remaining treatments (Control: 0 s, Z10: 0.4 ± 0.6 s, Z20: 26 ± 64.3 s, Z40: 63 ± 136.3 s, Z80: 130 ± 171.9 s, Z100: 245 ± 273.0 s; Appendix A). Similarly, immobility was rarely observed in mice in the Control (Figure 1d), but it was observed in the dosed treatments, especially in mice in Z100 (Control: 0.1 ± 0.6 s, Z10: 24 ± 43.4 s, Z20: 40 ± 56.9 s, Z40: 68 ± 122.9 s, Z80: 64 ± 76.4 s, Z100: 162 ± 118.1 s; Appendix A).

Recumbency and shaking (Figure 1e,i, respectively) were not observed for any Control mice and for less than 9 s in Z10, Z20, and Z40 treatments; some mice in Z100 treatment spent time expressing these behaviours (seven mice were recumbent for 84 ± 94.6 s, and four mice were shaking for 14 ± 20.7 s; Appendix A), and four mice in Z80 treatment spent almost all of the 800 s observation time being recumbent and shaking (343 ± 357 s, and 358 ± 378.2 s, respectively; Appendix A).

### 3.3. Euthanasia Period

The frequency of rearing decreased as the dose of tiletamine-zolazepam increased (Figure 2). More than 50% of the mice that were offered tiletamine-zolazepam doses up to and including 20 mg/kg (Control, and Z10 and Z20 treatments) reared at least one time during the euthanasia period (Figure 2). On the other hand, less than 20% of mice in both Z80 and Z100 treatments reared. Vertical jumps were only observed in Control and Z10 treatment, where 7.5% (*n* = 3) and 12.5% (*n* = 1) of mice, respectively, jumped one time or more during the euthanasia period. Jumping was not observed for any of the other treatments.

The behaviours recorded during the euthanasia period for the actual doses of tiletamine-zolazepam are presented in Figure 3. In general, mice that received more than 20 mg/kg of tiletamine-zolazepam presented high behavioural variability, especially regarding latency to ataxia from CO_2_ (Figure 3a) and latency to recumbency from ataxia (Figure 3b). Control mice, and those in Z10 and Z20 treatments, started presenting ataxia after 16 s of being exposed to CO_2_, while it took ≤11 s for mice in Z40 and Z80, and less than 7 s for those in Z100 (Appendix A). In contrast, mice in Control and Z10 and Z20 treatments were recumbent after no longer than 10 s of presenting ataxia, while it took 11 s for those mice in Z80 group, and nearly 15 s for those in Z40 and Z100 treatments (Appendix A). However, when latency to recumbency was considered from the onset of CO_2_ instead of from the onset of ataxia (Figure 4), mice that received doses of tiletamine-zolazepam between 80 and 100 mg/kg numerically took less time to become recumbent. The latencies for the other behaviours followed the same pattern when reported either from the onset of CO_2_ or the onset of the previous behaviour. 

Laboured breathing was observed right after recumbency, and in some cases even before (Control and Z80 treatment; Appendix A). On average, most mice started showing laboured breathing 1 to 2 s after being recumbent (Figure 3c); however it took 3 s and 0.5 s for mice in Z20 and Z80 groups, respectively.

The time from laboured breathing to last breath was 70 ± 7.3 s and 71 ± 10.7 s, for Control and Z10 treatment, respectively; while it took 92 ± 9.4 s for mice in Z100 treatment, and approximately 80 s for the rest of the treatments (Figure 3d; Appendix A).

## 4. Discussion

Euthanasia is the provision of a good death, and should ensure minimal discomfort, pain or distress for welfare, ethical and legal reasons [28]. Despite this, there is no definitive guidance on whether and how CO_2_ can be administered humanely [28]. Following the NC3Rs guidelines, refinement methods should be pursued to minimise animal suffering and improve welfare [29]. Thus, until a better alternative is identified, efforts should be made in finding refinement methods to improve CO_2_ euthanasia in rodents. In this study, we tested the voluntary ingestion of a sedative with cream cheese as a vehicle food prior to euthanasia with CO_2_ in mice. While mice can be neophobic [19,20], in our study all but two mice readily consumed the cream cheese after 3 days of habituation. We suggest this voluntary oral method is an affordable and easy method for administration of drugs; it is a refinement over other drug delivery options, as voluntary ingestion is a less invasive method than injection or gavage [19,20]. Nonetheless, while no mouse refused to consume their dosed cream cheese, tiletamine-zolazepam dosages of 40 mg/kg, 80 mg/kg, and 100 mg/kg resulted in some mice failing to consume their full amount. This was more evident in males, which consumed less of their dosed cream cheese than females. Teixeira-Santos et al. [19] explored oral drug administration by voluntary intake in mice and reported that females took longer to ingest the entire volume offered. The difference between our study and Teixeira-Santos et al. [19] study could be explained because in the latter, mice were not only habituated to the vehicle food but also to the drug and vehicle food combination; we only presented the drug on euthanasia day. Unfortunately, we were not able to confirm whether the change in consumption behaviour (from habituation to euthanasia day) in our study was related to changes in flavour, scent or other characteristics related to the tiletamine-zolazepam, or if onset of sedation prevented mice from consuming more. Cream cheese was chosen due to availability, palatability and because it was easy to mix with the sedative; however, a sweeter vehicle food, such as strawberry jam [19] could have hidden any possible changes in flavour due to the sedative more than cream cheese did. Our results may suggest that an acute exposure to tiletamine-zolazepam does not have rewarding and reinforcing effects, which is in accordance with what has been reported in rats [30].

During the sedation period, we found a reduction of rearing and lid walking, and an increase of the duration of ataxia, immobility and recumbency, with higher tiletamine-zolazepam dosage. Rearing has been used as a measure of exploration and anxiety in mice [31,32,33]; thus, a decrease in the frequency of rearing could be interpreted as a sign of reduced exploratory and anxiety-like behaviour; however, these latter studies recorded exploration and anxiety measures during specific tests, such as the open field or staircase test, and may not be representative of what happens in less disturbed conditions (e.g., when mice are in their home-cage as in our study). The decreased rearing could be a sign of muscle weakness, indicative of sedation [11]. In our study, this could explain not only the lower rearing frequency, but also the low lid-walking durations observed in mice receiving the highest doses (80 mg/kg and 100 mg/kg).

The presence of ataxia in mice in all treatments except Z10 (10 mg/kg) during the sedation process is likely to be associated with the effects of tiletamine-zolazepam, as it has been reported that diazepam (a benzodiazepine) can induce ataxia in rodents with intraperitoneal doses of 10 and 20 mg/kg [34]. Interestingly, not all mice responded consistently; some mice which consumed a low dose of tiletamine-zolazepam (i.e., <40 mg/kg) had high (>200 s) ataxia durations, while some mice which consumed a high actual dose (i.e., <70 mg/kg) displayed minimal ataxia (<50 s). Other behaviours, including increased duration of immobility and recumbency followed a similar, although less pronounced, pattern; such increases all are suggestive of sedation. While we had also anticipated increased shaking, paddling and laboured breathing, we did not observe this in our study (except for four individuals consuming between 50 and 70 mg/kg of tiletamine-zolazepam presenting high shaking duration). It has been described that methods for drug delivery through voluntary intake may be affected by sex, strain, and genetic factors [19]. In our study, all mice were the same strain and of similar age, but males weighed on average 5.9 ± 1.9 g more than females. On euthanasia day, food and water were removed from the cages before providing the dosed cream cheese, but differences in the ingestion of food prior the administration of the sedative could have affected the pharmacokinetics of the drug [35,36] and could explain the variability among individuals.

Overall, reduced rearing and lid-walking, and the presence of some ataxia, immobility and recumbency, all suggest that mice receiving the 20 mg/kg dose experienced mild sedation; these mice consumed their total amount of cream cheese, and thus received a consistent tiletamine-zolazepam dose. While higher dose treatments (40 to 100 mg/kg) produced more pronounced indicators of sedation, many mice did not consume their total amount, suggesting that higher doses may reduce the consistency of this delivery method to provide the intended dose. Therefore, we suggest that the 20 mg/kg of tiletamine-zolazepam could be a suitable oral dose for mice; however, this will need to be confirmed with dose–response studies.

During the euthanasia period, we anticipated that behaviours associated with CO_2_ aversion, such as rearing and jumping events, and latency to onset of ataxia, recumbency and last breath, would all be reduced with increasing tiletamine-zolazepam dosage. Spiacci et al. [14] reported that CO_2_ exposure evokes an active escape response in mice, indicating that reduced rearing in the euthanasia period would be a good indicator of reduced aversiveness. We observed mice rearing less with higher tiletamine-zolazepam doses; indeed, many mice consuming high doses of tiletamine-zolazepam (Z80 and Z100 treatments) never reared. Nonetheless, we caution that the lack of rearing could have been an inability to rear, as opposed to reduced aversiveness, either from tiletamine-zolazepam sedation or from the CO_2_; Marquardt et al. [11] suggested that reduced rearing was indicative of muscle weakness from an early-stage narcosis related to CO_2_. Therefore, it is important that if sedation is used in conjunction with CO_2_ in future work, effort is placed on quantifying the actual experiences created by the sedation versus the CO_2_ itself. As with rearing, vertical jumps are also interpreted as an escape response during hypoxia and used as a marker of aversion [24,37]; however, vertical jumps were not observed for any mice receiving 20 mg/kg or higher dose of tiletamine-zolazepam, in both the sedation and the euthanasia periods; therefore, it is likely that the mice were sedated enough for this behaviour to be prevented during CO_2_ exposure. Spiacci et al. [14,37] reported a similar decrease during hypoxia after the treatment with the benzodiazepine, alprazolam. Nonetheless, as with the lack of rearing, it is not possible to discern whether the reduced jumping was caused by a physical inability to jump, or whether the aversiveness of CO_2_ was actually reduced. CO_2_ concentration in our study was the same across treatments, and similar results were observed in Control and mice in Z10 treatment regarding rearing and jumping frequency. Améndola et al. [38] suggested that the aversion to CO_2_ in rats is driven by feelings of anxiety. Therefore, the lower rearing and jumping frequency observed for mice that received ≥20 mg/kg of tiletamine-zolazepam in comparison to those that received ≤10 mg/kg, could suggest less aversion to CO_2_ due to a reduction of the anxiety to this gas. This is in accordance with other authors who reported that benzodiazepines reduce the anxiety to CO_2_ in both mice and rats [38,39]. It is important to note that time between the sedation and euthanasia periods was not standardised. However, those mice that showed signs of sedation during the sedation period continued showing these signs at the beginning of the euthanasia period; suggesting that the effect of tiletamine-zolazepam did not wear off during this period. Saha et al. [40] reported that rats that received tiletamine-zolazepam showed a dose-dependent increase in duration of anaesthesia, being possible that mice that received high doses of the sedative could have experienced sedation for a longer time.

From the onset of CO_2_, tiletamine-zolazepam reduced the latency to ataxia; mice that received the highest dose (Z100 treatment) took less than 7 s to show signs of ataxia (e.g., loss of balance, gait irregularity, loss of muscle strength), while mice that received doses between 0 and 20 mg/kg (Control, and Z10 and Z20 treatments) started showing ataxia after 16 to 17 s of being exposed to CO_2_. Conversely, the time to recumbency from ataxia followed an opposite pattern (i.e., mice in Z100 treatment took nearly 15 s to be recumbent from the onset of ataxia, while Control mice and those in the Z10 and Z20 treatments took 7 to 9 s). This shows that a higher tiletamine-zolazepam dose made mice ataxic quicker and for longer, than mice which consumed lower doses. Ataxia has been associated with the anesthetic effects of CO_2_ and used as a marker of potential distress, but it is not clear whether it is a distressing experience or not [41]. It is likely that the differences in the onset of ataxia observed in this study were related to the effects of tiletamine-zolazepam as the concentration of CO_2_ was the same across treatments.

The onset of recumbency is an important measure as it has been associated with unconsciousness; Coenen et al. [42] supported this with the onset of an abnormal electroencephalogram pattern, as did Boivin et al. [41], who showed decreases in the blood pressure and heart rate curves occurring with ataxia. Moody et al. [25] described the onset of recumbency as the first and easiest indicator to identify loss of sensibility during euthanasia; these authors also described the loss of the righting reflex, and loss of the pedal withdrawal reflex as the second and third progressive measures to indicate loss of consciousness. Hickman et al. [43] used the point when a rat either touched its nose to the ground or began staggering with significant ataxia (gross loss of motor function) to determine loss of consciousness. However, all authors recognized that these measures may not accurately represent the onset of unconsciousness of the animals. Therefore, considering the latency to recumbency from the onset of CO_2_ as an indicator of loss of consciousness, mice in our study that received 80 and 100 mg/kg of tiletamine-zolazepam (Z80 and Z100 treatments) took less time to become recumbent than mice in Control and Z10, Z20 and Z40 treatments. Thus, it is likely that mice pre-treated with doses of tiletamine-zolazepam of ≥80 mg/kg achieved unconsciousness faster, and therefore could have experienced the negative effects of CO_2_ for shorter periods of time. However, variability in the onset of recumbency in the groups that received the higher doses was likely related to the incomplete intake of tiletamine-zolazepam which made the interpretation of the results difficult. Four mice in Z80 treatment and two mice in Z100 treatment were already recumbent before euthanasia and therefore, they could not be included in the euthanasia behavioural results; as mentioned previously, we cannot be certain how these mice were experiencing the sedative effects of tiletamine-zolazepam, but arguably entering the CO_2_ phase of the study while already recumbent, would be less aversive than what was experienced by the mice which took more than 23 s to become recumbent. Silverman et al. [44] reported that tiletamine-zolazepam induced anaesthesia without analgesia in mice at doses of 80 mg/kg or higher, and doses between 100 and 160 mg/kg caused respiratory distress and even death; however, it is important to mention that a small number of mice were used in Silverman’s study, and therefore we caution conclusions being made from this early work. Regardless, we suggest this is why tiletamine-zolazepam is not recommended as an anesthetic agent alone in mice, and it is usually combined with other drugs for this purpose [16]. However, as the aim of our study was to achieve sedation prior to euthanasia, the use of other drugs in combination with tiletamine-zolazepam was not considered.

The onset to laboured breathing from recumbency did not seem to follow any pattern with increasing doses of tiletamine-zolazepam, but the latency to last breath from the onset of laboured breathing did; this shows that in general, as the dose of tiletamine-zolazepam was higher, mice tended to have their last breath later, and therefore presented laboured breathing for longer. Laboured breathing is likely related to the effects of CO_2_ [7], and it has been suggested that might be distressing in mice as it is in humans [43]. As mentioned before, it is likely that mice that received doses of tiletamine-zolazepam ≥ 20 mg/kg could have experienced less aversiveness to CO_2_. However, we cannot confirm if laboured breathing was exclusively related to the CO_2_ or if it was also affected by tiletamine-zolazepam, as doses ≥100 mg/kg have been associated with respiratory depression [44]. Time to death is not easy to determine. Last breath could indicate time to death, but other measures would be needed to confirm this. For example, Smith and Harrap [45] considered that a rat was dead when its blood pressure was below 20 mmHg, which happened approximately 0.5 s after the cessation of breathing, and Boivin et al. [41] determined death in mice with the cessation of heartbeat. Boivin et al. [41] also reported that the unequivocal time point of death occurred between 69 and 204 s after the onset of CO_2_, the number depending on the chamber replacement rate.

It is important to mention that our study had some limitations. We therefore caution the reader to consider these when interpreting the results. First, due to resourcing limitations, we were unable to standardise the time between the sedation and euthanasia periods (which ranged from 2 to 24 min). As mentioned, it is possible that these inconsistencies resulted in some of the variability we observed between and within treatments. Second, a camera failure resulted in an inability to provide details regarding the 5 min of cream cheese intake prior to the sedation period; latency to first eat the cream cheese, total time to ingest it, eating bouts, and other behaviours such as escape attempts, could give additional information regarding the incomplete intake of the dosed cream cheese found when mice were provided 40, 80 and 100 mg/kg of tiletamine-zolazepam. Third, behavioural changes were the only measure used to assess distress during euthanasia with CO_2_ in this pilot study. The assessment of unconsciousness would need to be confirmed with physiological outcomes such as the loss of righting reflex, heart rate or blood pressure changes [11,41]; however, specific behaviours have been considered as good indicators of pain or distress during euthanasia with CO_2_ [11,12,41], and their measurement does not affect the animal nor the procedure. Finally, difficulty in standardising how time of death was determined from behavioural measures alone meant we could not report the latency to this variable, and also resulted in inconsistent length of time that the CO_2_ was on (i.e., it ranged from 1.4 to 2.3 min). We highlight these shortcomings so that future work could be planned to avoid them; nonetheless, we suggest that the results of the current study show merit for using voluntary oral ingestion as a dosing method and may also be indicative that the aversiveness of CO_2_ could be reduced using this method to administer a sedative.

## 5. Conclusions

This study gives new insights into the voluntary administration of a sedative (tiletamine-zolazepam) to mice prior to euthanasia with CO_2_. This could be a refining method to minimize distress and aversiveness during euthanasia with this gas. A dose of 20 mg/kg of tiletamine-zolazepam was found to achieve mild sedation prior to euthanasia and reduced aversive behaviours during euthanasia with CO_2_. Higher tiletamine-zolazepam doses (40 to 100 mg/kg) were also associated with signs of sedation (moderate to severe), however, the low incidence of escaping behaviours (rearing and jumping) in mice that received these doses could be either due to a reduced aversiveness to CO_2_ (less anxiety) or to the prevention of performing these behaviours. Furthermore, tiletamine-zolazepam doses between 40 to 100 mg/kg resulted in an incomplete intake of the cream cheese, and therefore, the real doses were lower than planned, especially for mice that received 80 mg/kg and 100 mg/kg.

We recommend that physiological measures are needed to determine the most appropriate dose of tiletamine-zolazepam to produce mild sedation in mice; this is particularly important because we found that doses higher than 20 mg/kg resulted in incomplete consumption of the allocated cream cheese by some mice. While we suggest that sedation achieved via voluntary ingestion of a drug, could be paired with CO_2_ to reduce its aversiveness during euthanasia, the limitations of our study must first be addressed; two important aspects we could not capture were determining the experience of the mice during sedation, as well as confirming unconsciousness and exact time to death.

In addition, we suggest that voluntary oral administration of cream cheese could be an effective, affordable, and easy way to minimize any handling stress, and thus could also be potentially used when administering other drugs to mice.

## Figures and Tables

**Figure 1 animals-11-02879-f001:**
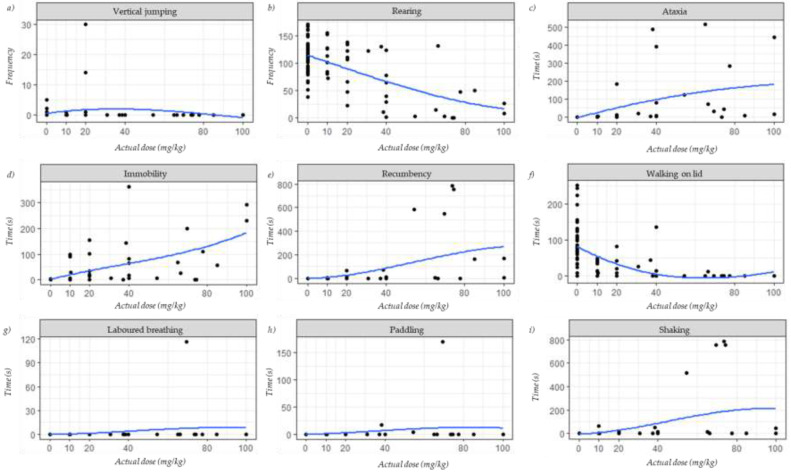
Behaviours reported for the first 800 s of the sedation period presented according to the actual doses of tiletamine-zolazepam in C57BL/6 mice: (**a**) vertical jumping; (**b**) rearing; (**c**) ataxia; (**d**) immobility; (**e**) recumbency; (**f**) walking on lid; (**g**) laboured breathing; (**h**) paddling; (**i**) shaking. Each point represents a single mouse. Blue trend lines are locally weighted smoothing (loess) lines.

**Figure 2 animals-11-02879-f002:**
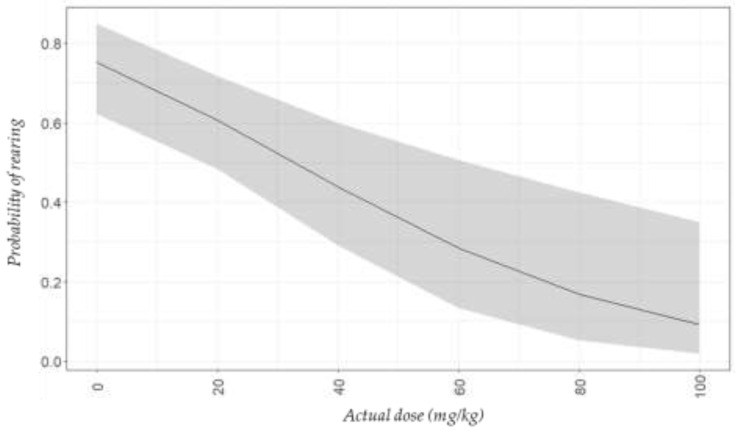
Predicted plot (±95% confidence interval) for probability of rearing (indicative of aversion) during euthanasia with CO_2_ according to the actual doses of tiletamine-zolazepam in C57BL/6 mice.

**Figure 3 animals-11-02879-f003:**
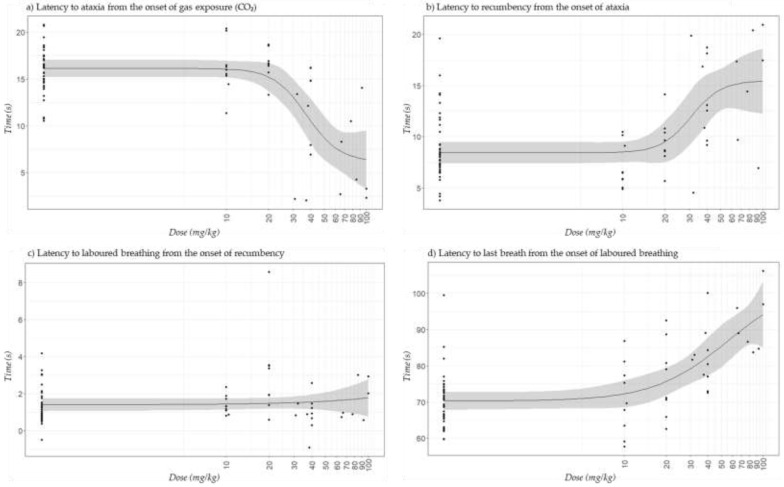
Predicted plots (±95% confidence interval) for the different behaviours observed during euthanasia with CO_2_ for the actual doses of the sedative tiletamine-zolazepam in C57BL/6 mice: (**a**) latency to ataxia from the onset of gas exposure (CO_2_); (**b**) latency to recumbency from the onset of ataxia; (**c**) latency to laboured breathing from the onset of recumbency; (**d**) latency to last breath from the onset of laboured breathing. Mice were observed until last breath (1.67 ± 0.31 min; mean ± SD). Each dot represents an individual mouse.

**Figure 4 animals-11-02879-f004:**
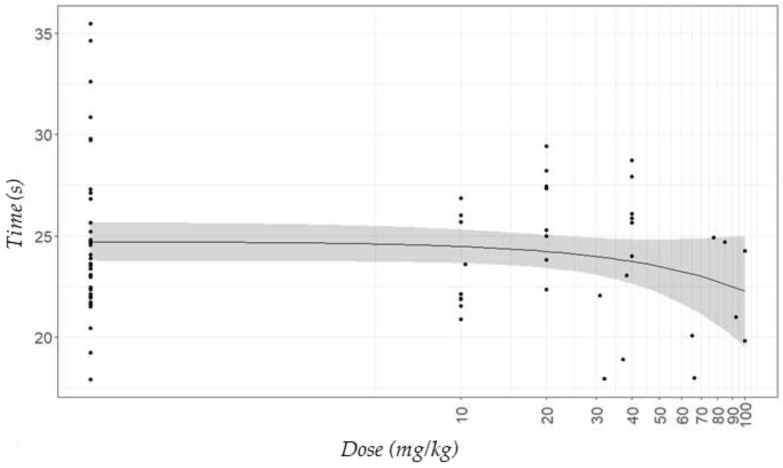
Predicted plot (±95% confidence interval) for the latency to recumbency from the onset gas exposure (CO_2_) during euthanasia with this gas considering the actual doses of the sedative tiletamine-zolazepam in C57BL/6 mice. Each dot represents an individual mouse.

**Table 1 animals-11-02879-t001:** Ethogram for coding sedation and aversive behaviours during sedation (pre-euthanasia) and euthanasia periods in C57BL/6 mice.

Behaviour	Definition	Sedation Period	Euthanasia Period
Ataxia	Presence of abnormal, uncoordinated, staggering movements [23,24]. Measured as duration(s) per mouse (in the euthanasia period time of onset until time of recumbency).	√	√
Recumbency	Head resting on cage floor, head and body motionless [25]. Measured as duration(s) per mouse (in the euthanasia period time of onset until time of laboured breathing).	√	√
Laboured breathing	Difficult breathing, with increased abdominal effort [24]. Measured as duration(s) per mouse (in the euthanasia period time of onset until time of last breath).	√	√
Jump	Rapid upward movement toward the top of the cage with all 4 feet leaving the floor [18,19]. Measured as frequency (total number of incidents per mouse).	√	√
Rearing	Up on hind limbs, with forefeet off the floor [21]. Measured as frequency (total number of incidents per mouse).	√	√
Immobility	Immobile state without any movements except respiration whilst the animal is standing with the head raised [12]. Measured as duration(s) per mouse.	√	
Walking on lid	Moving along the same route on the cage lid with all four legs. Measured as duration(s) per mouse.	√	
Shaking	Involuntary, somewhat rhythmic, muscle contraction and relaxation involving oscillations or twitching movements of the legs. Used once animals were sedated. Measured as duration(s) per mouse.	√	
Paddling	Multiple or single limbs moving in a way that resembles walking when the mouse is recumbent. Involuntary. Suggestive of an excitatory/light stage of anaesthesia. Measured as duration(s) per mouse.	√	

**Table 2 animals-11-02879-t002:** Planned and actual doses (mg/kg) of tiletamine-zolazepam for the different treatments in pair-housed C57BL/6 mice.

Treatment	Planned Dose	Actual Dose ^1^				*n* ^2^
		Mean	Min	Max	SD	
Control	0.0	0.0	0.0	0.0	0.0	40
Z10	10.0	10.0	10.0	10.0	0.0	8
Z20	20.0	20.0	20.0	20.0	0.0	8
Z40	40.0	38.5	30.8	40.0	3.3	8
Z80	80.0	58.1	10.4	77.6	23.2	8
Z100	100.0	81.9	32.0	100.0	23.0	8

^1^ As dosage increased, mice did not always consume all cream cheese offered, reducing the amount of tiletamine-zolazepam (Zoletil^®^) consumed. ^2^ Each treatment contained 2 cages of female, pair-housed mice and 2 cages of male, pair-housed mice; each of these was matched with a randomly assigned control, resulting in a total of 8 control mice per treatment, and a total of 40 control mice for the entire study.

## Data Availability

The data presented in this study will be openly available in Mendeley Data on the 21st of August 2022: https://data.mendeley.com/datasets/zzzvwpzyvt/1 (accessed on 1 August 2021). Rodriguez-Sanchez, Raquel; Barnaby, Elyssa; Amendola, Lucia; Hea, Shen; Smith, Bobby; Webster, James; Zobel, Gosia. 2021. Voluntarily ingestion of a sedative prior to euthanasia with CO_2_ in mice, Mendeley Data, V1, doi: 10.17632/zzzvwpzyvt.1.

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
