# Peer review of "Voluntary Oral Ingestion of a Sedative Prior to Euthanasia with CO2: Behavioural Responses of Mice"

_animals, 2021, doi:10.3390/ani11102879_

Round 1

Reviewer 1 Report

Please see file. I do think this should be published but I think it needs to be clarified a little. There needs to be a bit more info about the pharmacokinetics of Zoletil in here, including discussion of its previous use in mice.

Reviewer 2 Report

This is a useful study and a clearly written paper. The results are difficult to interpret without probabilities attached. Specific comments follow:

Title seems like it should read ' Voluntary...' (not 'voluntarily') but this is used throughout the text. Is this to distinguish it as 'voluntarily consumed' as opposed to gavage which is not voluntary?

L175 Spelling 'differed'

3.2 and 3.3 I'm not clear why the same behaviours (eg rearing) are described and treated differently for summary statistics in these sections - eg number of rears vs frequency of rearing in the sedation vs euthanasia periods, respectively.

Line 283-284 Do you mean that rearing and lid walking are reduced during sedation? This appears to say that these behaviours indicate sedation (as opposed to a reduction in these behaviours indicates sedation).

L308 'weighed' (not 'weighted')

L387 'incomplete' (not 'uncomplete')

L407 'mentioned' (not 'mention')

L429 'physiological' (not 'physiologic')

L441 'voluntary administration' not 'voluntarily administration'
